# Community Initiatives to Promote Basic Life Support Implementation—A Scoping Review

**DOI:** 10.3390/jcm10245719

**Published:** 2021-12-07

**Authors:** Andrea Scapigliati, Drieda Zace, Tasuku Matsuyama, Luca Pisapia, Michela Saviani, Federico Semeraro, Giuseppe Ristagno, Patrizia Laurenti, Janet E. Bray, Robert Greif

**Affiliations:** 1Institute of Anaesthesia and Intensive Care, Catholic University of the Sacred Heart, Fondazione Policlinico Universitario A. Gemelli, IRCCS, 00168 Rome, Italy; luca.pisapia92@gmail.com (L.P.); saviani.michela@gmail.com (M.S.); 2Italian Resuscitation Council, 40128 Bologna, Italy; drieda.zace@unicatt.it (D.Z.); federicofsemeraro@gmail.com (F.S.); giuseppe.ristagno@unimi.it (G.R.); 3Section of Hygiene, University Department of Life Sciences and Public Health, Catholic University of the Sacred Heart, 00168 Rome, Italy; patrizia.laurenti@unicatt.it; 4Department of Emergency Medicine, Kyoto Prefectural University of Medicine, Kyoto 602-8566, Japan; task-m@koto.kpu-m.ac.jp; 5Department of Anaesthesia and Intensive Care and EMS, Maggiore Hospital Bologna, 40133 Bologna, Italy; 6Department of Pathophysiology and Transplantation, University of Milan, 20122 Milan, Italy; 7Department of Anesthesiology, Intensive Care and Emergency, Fondazione IRCCS Ca’ Granda Ospedale Maggiore Policlinico, 20122 Milan, Italy; 8Department of Epidemiology and Preventive Medicine, Monash University, Melbourne, VIC 3400, Australia; janet.bray@monash.edu; 9Department of Anaesthesiology and Pain Medicine, Bern University Hospital, University of Bern, 3010 Bern, Switzerland; robert.greif@insel.ch; 10School of Medicine, Sigmund Freud University Vienna, 1020 Vienna, Austria

**Keywords:** basic life support, community initiatives, outcome, out-of-hospital cardiac arrest, scoping review, cardiopulmonary resuscitation, bystander cardiopulmonary resuscitation

## Abstract

Introduction: Early intervention of bystanders (the first links of the chain of survival) have been shown to improve survival and good neurological outcomes of patients suffering out-of-hospital cardiac arrest (OHCA). Many initiatives have been implemented to increase the engagement of communities in early basic life support (BLS) and cardiopulmonary resuscitation (CPR), especially of lay people with no duty to respond. A better knowledge of the most effective initiatives might help improve survival and health system organization. Aim of the scoping review: To assess the impact of specific interventions involving lay communities on bystander BLS rates and other consistent clinical outcomes, and to identify relevant knowledge gaps. Methods: This scoping review was part of the continuous evidence evaluation process of the International Liaison Committee on Resuscitation (ILCOR), and was performed following the Preferred Reporting Items for Systematic reviews and Meta-Analyses extension for Scoping Reviews. We performed a literature search using the PubMed, EMBASE, and Cochrane databases until 1 February 2021. The screening process was conducted based on predefined inclusion/exclusion criteria, and for each included study, we performed data extraction focusing on the type of intervention implemented, and the impact of these interventions on the specific OHCAs outcomes. Results: Our search strategy identified 19 eligible studies, originating mainly from the USA (47.4%) and Denmark (21%). The type of intervention included in 57.9% of cases was a community CPR training program, in 36.8% bundled interventions, and in 5.3% mass-media campaigns. The most commonly reported outcome for OHCAs was bystander CPR rate (94.7%), followed by survival to hospital discharge (36.8%), proportion of people trained (31.6%), survival to hospital discharge with good neurological outcome (21%), and Return of Spontaneous Circulation (10.5%). Community training programs and bundled interventions improved bystander CPR in most of the included studies. Conclusion: Based on the results of our scoping review, we identified the potential benefit of community initiatives, such as community training in BLS, even as part of bundled intervention, in order to improve bystander CPR rates and patient outcomes.

## 1. Introduction

In out-of-hospital cardiac arrest (OHCA), the first three links of the chain of survival [1] are referred to as basic life support (BLS), and include early recognition of cardiac arrest, calling the local emergency service [2], providing bystander cardiopulmonary resuscitation (CPR, i.e., chest compressions with or without rescue ventilations) [3], and retrieval and use of an automated external defibrillator (AED) [4]. When applied rapidly, these BLS interventions offer the greatest chance of OHCA survival and good neurological outcomes [5]. However, the delivery of these interventions is still far from optimal, with only ~80% of OHCA recognized, and large regional variation in rates of bystander CPR and AED use [6,7,8]. Over the past two decades, many interventions aiming to improve the engagement of lay people with no formal duty to respond to OHCA have been tested and are now recommended in international guidelines. Examples include dispatcher-assisted CPR, public access defibrillation (PAD) programs, and AED dissemination including drones’ deployment, simplification of CPR (i.e., chest compressions only CPR), and apps to localize and engage first responders and/or the nearest AED [9]. What is less understood is the impact of other community-based initiatives aiming to improve CO-CPR rates, especially those which promote or provide public BLS education and training [9].

In 2020, an evidence review, to obtain a better understanding of the effectiveness of community based initiatives in improving BLS implementation, was prioritized by the Education, Implementation and Teams (EIT) Task Force for the International Liaison Committee on Resuscitation. An initial literature review showed high heterogeneity between studies, and did not reveal sufficient comparative studies to justify a systematic review with a meta-analysis. Therefore, the Task Force decided to conduct a scoping review providing a broader overview of the community-based strategies used to promote BLS and additionally to identify relevant knowledge gaps. 

## 2. Methods

### 2.1. Protocol

This scoping review was conducted according to the ILCOR processes for scoping reviews [10], and followed a recommended methodological framework [11] according to the Preferred Reporting Items for Systematic reviews and Meta-Analyses extension for Scoping Reviews (PRISMA) [12]. 

### 2.2. Objectives

We searched for studies investigating interventions aimed to improve the actual implementation of BLS in communities. We excluded simulation studies or those overlapping with other ILCOR reviews.

### 2.3. PICOST Definition

The following PICOST (Population, Intervention, Comparator, Outcome, Study Designs and Timeframe) question was defined a priori:Population: within the general population of children and adults suffering an out-of-hospital cardiac arrest (OHCA);Intervention: do community initiatives promoting Basic Life Support (BLS);Comparison: in comparison to current practice;Outcomes: have any impact on:(1)the survival to hospital discharge with good neurological outcome;(2)survival to hospital discharge;(3)return of spontaneous circulation (ROSC);(4)time to first compression;(5)bystander CPR rates; or(6)proportions of the population trained in BLS.Study Designs: Randomized controlled trials (RCTs) and non-randomized studies (non-randomized controlled trials, interrupted time series, controlled before-and-after studies, cohort studies) were eligible for inclusion;Time: No limit.

### 2.4. Search Strategy

We searched three major databases (PubMed, EMBASE, and Cochrane; DZ developed the search strategy), looking for relevant articles published until 1 February 2021. The search strategies were first drafted and further refined through a Task Force discussion. The keywords were: “Out-of-Hospital Cardiac Arrest”, “Heart Arrest”, “Cardiopulmonary Resuscitation”, “Basic Life Support”, “Lay People”, “First Responder”, “Bystander”, “Community-Based Initiative”, “Community Involvement”, “Public Engagement”, and “Community-Driven Intervention”, combined through the Boolean operators AND, OR. The search was restricted to only humans. With this exception, no other filters were used. The full search strategy for all the databases is available in Appendix A.

### 2.5. Selection Process

#### 2.5.1. Definitions

For the purpose of this review, we defined “community” as the general population of a defined geographical area (i.e., a group of neighborhoods, one or more cities/towns or regions, a part of or a whole country), in which individuals with no duty or organized role to respond can act as potential witnesses or bystanders of a OHCA victim. This definition does not include healthcare professionals or first responders with a role as rescuer or part of a response system. 

We defined “initiative” as any intervention promoting BLS or providing education and training, aiming to increase the engagement of the community (as defined above) in providing BLS (i.e., any kind of CPR and early defibrillation). 

#### 2.5.2. Inclusion Criteria

Studies were eligible for inclusion if they addressed the research question, reported the impact on rates of training or OHCA outcomes, and were published in English in peer-reviewed journals. 

#### 2.5.3. Exclusion Criteria

Studies were excluded that: did not address the research question; reported findings in only abstract form; overlapped with other topics already investigated in specific ILCOR reviews (e.g., PAD programs or other AED dissemination and deployment programs including use of drones; dispatcher and/or Telephone CPR; use of Apps for FR dispatch and/or AED localization; impact of social or economic factors in bystander’s engagement; effect of different CPR techniques or protocols including changes in resuscitation guidelines); and those examining healthcare professionals as individuals or part of medical systems (physicians, dentists, nurses, emergency medical technicians, pharmacists, and students), as well as those with a duty to respond as FR, such as lifeguards, firefighters, and police officers. Grey literature was not included.

#### 2.5.4. Study Selection 

Articles identified by the search strategy were imported to RAYYAN QCRI software [13], and duplicates were removed. The screening process was carried out separately by four reviewers (AS, JB, DZ, LP) working in pairs, and was divided in two rounds. During the first round, pertinent articles were selected based on titles/abstracts. Then, the full texts of these articles were obtained and entirely read, and those satisfying all the inclusion criteria were included in the review. Disagreements were resolved by team discussion. The reference lists of the included studies were hand searched to look for additional articles.

### 2.6. Data Extraction and Synthesis

A dedicated data extraction form was used to retrieve information for each eligible article (see Table 1). The narrative synthesis was performed based on the type of implemented initiative. For this purpose, we grouped the interventions found in the included studies in three categories: (1) Community Training Programs; (2) Mass-Media Campaign; and (3) Bundled Interventions. 

In the case of studies reporting the impact of bundled interventions, which included FR programs and EMS interventions, we only extracted outcome data relevant to the included initiative and not the combined intervention. For example, if a study implemented a mass media CPR campaign combined with a first responder program, we only examined the rate of bystander CPR (that is the effect of the mass media campaign, which we define as a community intervention) and not survival (that is affected by both the bystanders and the first responders CPR, the latter being an element of the response system). 

## 3. Results

### 3.1. Search Strategy

Our search strategy found a total number of 2656 articles. Figure 1 presents the results of the screening process, after which 19 studies [14,15,16,17,18,19,20,21,22,23,24,25,26,27,28,29,30,31,32] were included in this scoping review (Table 1).

### 3.2. Characteristics of the Included Studies

#### 3.2.1. Geography

The included studies (*n* = 19) were conducted in the USA (47.4%) [14,15,16,17,18,19,20,25,32], Denmark (21%) [21,22,23,26], Republic of Korea (21%) [27,28,29,31], Japan (5.3%) [30], and Singapore (5.3%) [24].

#### 3.2.2. Study Design

The majority of the included articles were cohort studies (47.4%) [15,16,17,21,22,23,26,30,32], followed by before-and-after studies (31.6%) [18,19,20,24,28,31], cross-sectional studies (10.5%) [27,29] RCTs (5.3%) [14], and one non-randomized controlled trial (5.3%) [25]. More than half of the studies had a prospective design (73.7%) [14,15,16,19,21,22,23,24,25,26,27,28,29,30], while the rest were retrospective (26.3%) [17,18,20,31,32].

#### 3.2.3. Time

Almost all studies were published during the last decade (2012–2019) (89.5%), with only two studies (10.5%) published earlier [14,25]. 

#### 3.2.4. Data Source

In five studies, data were obtained from the national registries: the Cardiac Arrest Registry to Enhance Survival (CARES), [17,18,20], and the Korean Community Health Survey (CHS) [27,29].

#### 3.2.5. Population

A total of 12 studies reported outcomes for more than one thousand OHCAs (63.2%) [15,16,17,20,23,24,25,26,27,29,31,32], three of which included more than ten thousand cases (15.8%) [26,27,29]. All OHCA cases were adult patients with a mean age between 59.8 ± 19.2 [20] and 74.7 ± 15.9 years old [30]. Males accounted for an overall 60% of cases. 

#### 3.2.6. Intervention Setting

The settings where the community interventions took place were workplaces, schools, governmental offices, major civic events, community-shared spaces, etc. Most OHCAs (from 68% in Fordyce et al. [16] to 87% in Uber et al. [20]) occurred in private places, commonly at home.

### 3.3. Description and Effectiveness of the Interventions

#### 3.3.1. Interventions

The type of intervention was: a community training program as a single intervention (±media campaigns) in 57.9% of cases (11 studies) [14,15,16,17,18,19,20,21,22,23,24], including instructor-led, peer-to-peer and/or self-training; a mass-media campaign in 5.3% (1 study) [25]; and bundled interventions (including combinations of mass-media campaigns, training and/or other measures with or without involvement of EMS, and/or in-hospital care organizations) in 36.8% (7 studies) [26,27,28,29,30,31,32]. 

#### 3.3.2. Outcomes

Reported outcomes of OHCAs included bystander CPR rates (94.7%), survival to hospital discharge (36.8%), the proportion of people trained (31.6%), survival to hospital discharge with good neurological outcome (21%), and ROSC (10.5%). Only a few studies reported adjusted outcomes (26.3%) [17,23,25,28,32]. No study reported time to first compressions (Table 2). 

1. Community Training Programs

There were eleven studies [14,15,16,17,18,19,20,21,22,23,24] that evaluated the impact of CPR training (±media) on outcomes after OHCA. In ten studies, the intervention included instructor-led training: it was the main intervention in five studies [15,16,18,20,24], and coupled with a self-learning video in three studies [19,21,22]). In one study [14], a 10-min videotape was the specific training tool. A total of two studies reported the role of peer-to-peer training [17,23]. In three studies, local media supported the initiatives with announcements and advertisement [17,21,22]. 

Survival to hospital discharge with good neurological outcome was reported in three studies using instructor-led training [15,16,20]. Only one study reported an overall improvement (7.1% to 9.7%, *p* = 0.02) [15]. Another study reported an improvement in OHCAs occurring in public places (9.5% to 14.7%, *p* = 0.02), but not in those occurring at home [16].

Survival to hospital discharge was reported in six studies [16,18,20,21,22,24]. One instructor-led training study reported an overall improvement (AOR 2.39 [1.02–5.62] *p* = 0.045) [24], and another reported an improvement in OHCA occurring both at home (5.7% to 8.1%, *p* = 0.047) and in public places (10.8% to 16.2%, *p* = 0.04) [16]. This outcome was not improved in the other four studies [18,20,21,22], including two with both instructor-led and self-learning training [21,22].

ROSC was reported by two studies [20,24], but only improved on one instructor-led study (5.1% to 9.1%, *p* = 0.01; AOR 1.94 [1.15–3.25]) [24].

Bystander CPR rate was reported in ten studies [14,15,16,17,18,20,21,22,23,24] and improvements were seen in seven studies [15,16,17,18,21,22,24]. Three of these studies examined instructor-led training, reporting increases of bystander CPR rate from 39.3% to 49.4% (*p* < 0.01) [15], from 83% to 95% [18] and from 44.8% to 63.7% (*p* < 0.001) [24], respectively. A forth instructor-led study reported increased bystander CPR rates in both home and public settings [16]. Two studies with instructor-led and self-learning training, described increases from 22% to 74% [21], 47% to 70% [22] and one peer-to-peer training study reported an incidence of bystander CPR 0.42 ± 0.34 vs. 0.47 ± 0.30, *p* < 0.05 [17]. Only three studies (one with self-learning and two with instructor-led training) [14,20,23] reported no improvement in bystander CPR rate.

None of the studies reported the time to first compression as an outcome.

2. Mass-Media campaigns

Only one study [25] explored the impact of mass media as the main initiative on bystander CPR rate. This study explored the impact of television public service announcements, reporting an increase in bystander CPR rate from 43% to 55% (*p* < 0.05) in the intervention group.

3. Bundled Interventions

Among the seven studies that assessed the impact of bundled interventions on OHCA outcomes, the main intervention was instructor-led training and course [26,27,28,29,30,31,32], together with sessions at public sites (2 studies [28,29], implementation of new guidelines (2 studies) [26,27], mandatory education at schools and courses when acquiring a driver license (1 study) [26], public campaigns (1 study) [29], or self-learning video and kit (1 study) [30].

Among these studies, bystander CPR rate was the most reported outcome (100%) [26,27,28,29,30,31,32], followed by proportion of population trained (28.5%) [28,30], and survival to hospital discharge (14.3%) and survival to hospital discharge with good neurological outcomes [30].

Mandatory BLS courses at schools or when acquiring a driver license together with new guidelines protocols increased bystander CPR in one study (21.1% to 40.9%, *p* < 0.001) [26]. The simultaneous implementation of instructor-led training and new guidelines on laypersons was reported to be beneficial for bystander CPR rate in one study [27]. Bystander CPR rate was also improved in one study, after instructor-led training including schools and workplaces [28]. A significant improvement in bystander CPR rate (60.1% to 638%, *p* < 0.001) was also reported as a consequence of instructor-led training and public CPR training in one study [29]. In total, three studies reported no significant improvement in bystander CPR rates after implementing a bundled intervention [30,32]. One study reported no improvement on survival to hospital discharge and survival to hospital discharge with good neurological outcomes [30].

## 4. Discussion

This review aimed to map and summarize the community initiatives to promote BLS, describe their impact in improving OHCA outcomes, and identify existing knowledge gaps.

In order to facilitate description and comparison between different interventions, we grouped the identified interventions into three main categories: community training programs, mass-media campaigns, and bundled interventions. According to these criteria, this scoping review found that the most commonly implemented intervention was instructor-led community training programs, while the most assessed outcome was bystander CPR rate.

### 4.1. Initiatives

#### 4.1.1. Community Training Programs

Regarding the studied initiatives, training by the mean of instructors has been widely used as an alone or combined intervention. Chest compression-only CPR has been the most frequent content of training, since it is a reasonable alternative to standard CPR (compressions plus ventilations) and can be easier to learn [33].

Instructor-led training can be considered the most common way to disseminate BLS skills. It is usually deployed during a BLS course, but instructors can teach BLS even in targeted contexts (public places, schools, families, workplaces) and in a shorter time. Instructor-led training was the only teaching intervention [15,16,18,20,24] or the first tier of a training program to start peer-to-peer training with the adjunct of self-learning videos [19,21,22].

Interestingly, when intervention training is delivered as a “single shot”, not in a program context and with no specific targets [19,20], or as time-limited initiative with a short follow up [23], the interventions do not seem to impact on outcomes. In general, all targeted and multiple sessions programs were able to increase at least bystander CPR.

When initiatives included schools or family members [15,16,24], survival increased as well as bystander CPR, with the exception of the study by Isbye et al. [23]. Furthermore, programs which engaged students as a second-tier trainers [19,23] with a self-learning kit were able to increase the proportion of trained people (with a ratio of 4.9 and 2.5 respectively).

#### 4.1.2. Mass Media Campaigns

Media campaigns as a sole initiative [25] or as an adjunct to community training programs [17,21,22] contributed to increase bystander CPR rates, but the evidence on survival is conflicting. Interestingly, despite mass media (television, radio, newspapers, and magazines) being considered the most obvious way to spread messages and promote change in behavior [34], our search was able to only find a small number of published studies evaluating their impact. Furthermore, we did not find studies including social media which have recently reached a prominent position in influencing opinions, attitudes, and practices. Focus-designed communication strategies have been successfully implemented in many areas of public health to reach targeted improvement and should be considered even in the field of bystander CPR [35,36]. A specific analysis of studies addressing social media as a tool of public campaigns could add useful information.

#### 4.1.3. Bundled Interventions

We found that bundled interventions, targeting different components of the “chain of survival”, can improve OHCA survival [26,27,28,29] probably better than isolated ones; however, it is difficult to isolate the effect of each single component of the program on outcomes. This aspect has been highlighted in a recent systematic review and meta-analysis [37], which investigated the effect of community initiatives on survival to discharge or 30-day survival and on bystander CPR rate, finding an improvement in both outcomes (OR, 1.34; 95% CI, 1.14–1.57; I2 = 33% and OR, 1.28; 95% CI, 1.06–1.54; I2 = 82%, respectively). In that review, the authors classified the interventions in community initiatives alone (five studies) and in initiatives that were combined with changes in health care services (ten studies). Community plus health service interventions were associated with a greater bystander CPR rate compared with community alone initiatives, while survival rate did not differ. However, many of the included studies involved non-health professionals with a duty to respond, such as firefighters and policemen, thus confounding the effect of interventions on their ability to engage occasional bystanders, intended as laypeople with no role in the emergency response systems. Interestingly, a restricted analysis on studies targeting laypeople only confirmed a positive association with increased bystander CPR. Nevertheless, in four out of nine of the studies included in this sub analysis, CPR training was associated with a notification system which, by definition, addresses laypeople who adhere to an organized response system on a volunteer basis, such as a “first responder system”. In contrast, the present scoping review excludes this kind of study, with the purpose of investigating and highlighting the effect of community initiatives on the willingness of general population to provide CPR at an earlier stage than the involvement in a first responders program.

### 4.2. Outcomes

With respect to outcomes, bystander CPR rate was reported in almost all the included studies, and, in most of them, it showed a benefit with the implementation of the intervention. This benefit was more frequent when the type of initiative was a ‘bundle’ of interventions compared to single training or mass-media initiatives.

There was insufficient evidence regarding the impact of community initiatives on other outcomes, such as survival with good neurological outcome, one-month survival, ROSC, and time to first compression, either because studies did not report them, or because these outcomes could be impacted by other initiatives that go beyond the objectives of this review. A benefit in survival at hospital discharge was reported in only 40% of studies that assessed this outcome. In these studies, it is difficult to establish an association between improved bystander CPR rate and improved survival, as the latter is probably multifactorial and most likely related to improvements in all the links of the chain of survival [15,16,24].

In the case of studies assessing bundled interventions, many outcomes were reported that could not be included in the narrative analysis, since it was impossible to isolate which specific intervention (respecting the inclusion and exclusion criteria) was associated with which outcome.

Therefore, since clinical outcomes such as ROSC or survival at any time can be affected substantially by patients’ conditions and emergency response system performance, we suggest considering bystander CPR rate as the most consistent and appropriate outcome to reflect the effectiveness of community initiatives as intended in the present review. In fact, bystander CPR rate is the only outcome completely dependent on laypeople willingness to intervene, and thus the more directly related with the implemented intervention.

However, it should be considered that community interventions to promote BLS and their effectiveness are context-specific and can be affected by individual characteristics, cultural sensitivity, medico-legal environment, and training method and quality. In fact, non-targeted approaches, and the training sites selection without consideration of the actual need for BLS learning in corresponding communities, have shown to lack efficacy [23]. Community initiatives aimed at public engagement demand multidisciplinary competencies in the project phase (such as experts in communication, social marketing, and public health programs) and significant investment of both human and financial resources for their deployment. Simplicity and brevity of the community training programs were important characteristics to allow for maximal efficiency and the training of individuals who do not wish to participate in a longer course or seek BLS certification [27]. These interventions may be particularly helpful when targeting areas with high OHCA incidence and low bystander CPR rates, which have also been shown to have low rates of CPR training [38], including lower socioeconomic status neighborhoods, since they can require lower costs and less time commitment [17]. In this context, BLS educational interventions addressing high-school students were identified as an important component, since they may disseminate CPR knowledge beyond the classroom, and reach into low-income, minority neighborhoods [19,23].

Public health initiatives to improve bystander CPR and early defibrillation are associated with better outcomes for OHCAs at home, where the prognosis has traditionally been poor [16]. Included studies underline the fact that multiple, multifaceted, and community-wide programs using training, media, advertisement materials, and public presentations may be needed to increase bystander CPR in the communities [26,27,28,29].

Based on the results of this scoping review and the narrative summary, implementation of community initiatives such as BLS training involving a large portion of population or bundled interventions can be considered to improve the bystander CPR rate among laypersons in cases of OHCAs. Furthermore, this scoping review provided information about different types of community initiatives that have been implemented in an attempt to improve bystander CPR. It might additionally offer suggestions to communities’ leaders, decision-makers, and other stakeholders to structure and implement policies and guidelines to engage laypersons in resuscitation to improve the outcomes of OHCA.

## 5. Knowledge Gaps

Despite not being a systematic review, this scoping review highlights important knowledge gaps, such as the need for more studies on this topic, more rigorously designed RCTs regarding this issue and studies on children. Furthermore, future studies should document outcomes such as survival with good neurological outcome, survival to discharge (hospital discharge and one-month survival), ROSC, and time to first compression. Specifically, we identified:

Data limited to only some geographical areas, which is a result of the published literature and a lack of evidence regarding community initiatives to promote BLS implementation in some countries;

A need for more high-quality studies, especially RCTs in order to have more robust evidence and outcomes adjusted for the main confounders;

A need to evaluate the effect of public campaigns (World Restart A Heart—WRAH), and BLS teaching at school (Kids Save Life—KSL)) on BLS training implementation, bystander CPR, and clinical outcome;

A need to evaluate the effect of specific legal regulations in different countries, which can facilitate BLS deployment of general population with the support of EMS-dispatchers;

No studies evaluated regional initiatives specifically for BLS in children or implementation programs including children in the chain of survival;

A need to isolate the specific interventions that are associated with the improvement or not of each specific outcome in studies assessing the impact of bundled interventions;

A need to investigate the cost-effectiveness of any single intervention and their specific impact on clinical outcomes.

## 6. Limitations

This work has some limitations. First it is a scoping review, thus it does not have the methodological rigorousness of a systematic review; and therefore, no recommendations regarding interventions are possible. As any review, it may be susceptible to a selection bias, since there may be articles that our search strategy might not have identified. Additionally, there are different definitions of “community initiatives” and therefore any review would face difficulties to get all relevant studies.

Moreover, many included articles reported limits in the conduct of the studies, such as missing data on CPR quality, as well as information on bystanders, including age, sex, occupation, and BLS training experience. Given the observational nature of most of the studies, causality cannot be determined, despite finding associations. Moreover, unmeasurable or unmeasured confounders could explain improved temporal outcomes independent of public health initiatives. In fact, only a few studies controlled for the main variables that could confound the association between community initiatives and OHCA outcomes. Finally, studies assessed the role of bundled interventions in an aggregate way, thus it was not possible to determine how much each single initiative contributed to the changes observed in the outcomes of interest.

## 7. Conclusions

This scoping review presents a narrative summary of various community initiatives that have been implemented in an effort to promote BLS. These included mass training across all age-groups or bundled interventions targeting different components of the “chain of survival” which had the potential to improve bystander CPR rate among laypersons in cases of out-of-hospital cardiac arrest.

## Figures and Tables

**Figure 1 jcm-10-05719-f001:**
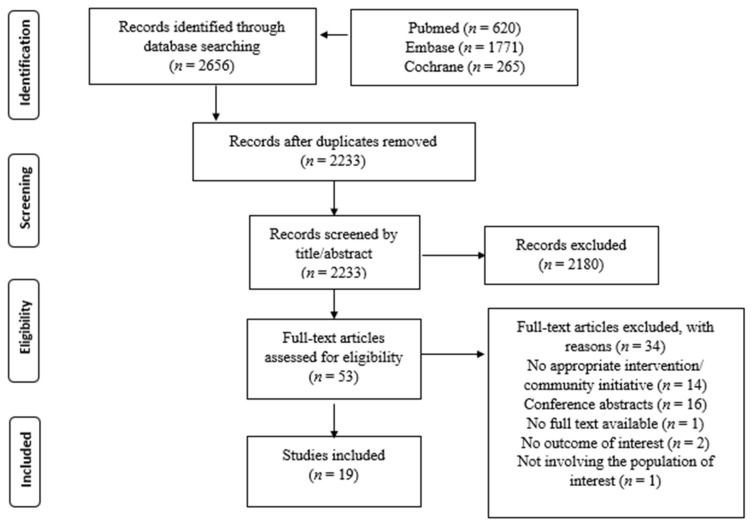
Flowchart of the screening process.

**Table 1 jcm-10-05719-t001:** Characteristics and outcomes of the included studies.

Author Year	Design Region (Country)	Population	Intervention(s)	Main Findings	Comments
**Community Training Programmes *n* = 11**					
Eisenberg 1995 [14]	Randomized controlled trial Washington (USA)	17,318 households: 8659 households intervention and 8659 household control. 65 OHCAs: 31 in intervention households and 34 in control households.	Self-training: via a free, mailed 10-min CPR training videotape, a brochure and pocket card illustrating CPR steps.	No impact of the intervention on rates of bystander CPR 47% vs. 53% (*p* = NS).	**Training** Self-training: videotape at home. Small number of OHCA events. Unknown reach of intervention
Hansen 2015 [15]	Prospective cohort study, North Carolina (USA)	Community members offered training. 4961 OHCAs	Instructor-led training offered at major civic events, in public places and to patients with cardiovascular disease and their family members. School staff were trained in the use of AED. Community grants provided to implement CPR training programs.	Associated increase rates of survival with favourable neurological outcome in patients who received CPR [7.1% (95% CI, 5.8–8.8%) in 2010 to 9.7% (95% CI, 8.2–11.4%) in 2013 (*p* = 0.02)] and increase in bystander CPR [39.3% (95% CI, 36.5–42.1%) in 2010 to 49.4% (95% CI, 46.7–52.0%) in 2013 (*p* < 0.01)], and defibrillation	**Training** Instructor-led in public places. Unable to isolate effect of training on outcomes. Unknown reach of intervention.
Fordyce 2017 [16]	Prospective cohort study North Carolina (USA)	Community members offered training. 8269 OHCAs: 5602 in homes and 2667 in public.	Instructor-led training offered at major civic events, in public places and to patients with cardiovascular disease and their family members. School staff were trained in the use of AEDs. Community grants provided to implement CPR training programs.	Increase in favourable neurological survival in OHCAs occurring in public (9.5% vs. 14.7% *p* = 0.02) but not at-home (4.9% vs. 6.1% *p* = 0.06). Improvement of survival to hospital discharge at home (5.7% vs. 8.1% *p* = 0.047) and in public (10.8% vs. 16.2% *p* = 0.04) Increase in bystander CPR in OHCAs in public (61% vs. 70.5%, *p* = 0.01) and in home (28.3% vs. 41.3%, *p* < 0.01).	**Training** Instructor-led in public places. Unable to isolate effect of training on outcomes. Unknown reach of intervention.
Bergamo 2016 [17]	Retrospective cohort study Texas, USA	Community residents 2474 OHCAs	Take 10 10-minute peer-to-peer training with promotion via word-of-mouth, media and calls to community organizations	1.09% (*n* = 11,242) of the population was trained Bystander CPR rates increased (Incidence 0.42 ± 0.34 vs. 2013: 0.47 ± 0.30; *p* < 0.05).	**Training** Peer-to-peer. Results are unadjustedfor any possible confounder. Unable to isolate effect of training on outcomes
Boland 2017 [18]	Retrospective before-after study Minnesota, USA	Community residents in areas of training 294 OHCAs	Heart Safe Communities. Community-specific action plans include educating citizens about the warning signs and symptoms of cardiac arrest, conducting training sessions on how to perform CPR and use AEDs, registering and mapping existing AEDs, and procuring and placing additional AEDs in strategic public locations	9% (*n* = 44,293) of the population was trained Bystander CPR rates increased [83% vs. 95% (OR = 4.23; CI 1.80–9.98)] No difference in survival to hospital discharge (17% vs. 20%, *p* = 0.32).	**Training** Community targeted. Results are unadjusted for any possible confounder.. Unable to isolate effect of training on outcomes.
Del Rios 2018 [19]	Prospective before-after study Chicago, USA	71 students and 347 friends and relatives	Instructor-led and self-training. Self: video/kit 1. Two in-class training sessions of 45 min each. 2. AHA CPR Anytime video self-instruction kit, including an instructional DVD and inflatable mannequins	Proportion of population trained: 71 students were trained for CPR, who later trained other 347 friends and family members. Proportion of population trained: 1: 4.9 people	**Training** Instructor-led and self-training with video. One shot initiative targeting schools using peer-to-peer techniques, DVD +kit. There was no CPR protocol reported.
Uber 2018 [20]	Retrospective before-after study Michigan, USA	1486 cardiac arrest patients (899 P1 and 587 P2). 2253 passers were trained.	Instructor-led, CO-CPR, 1 day training in public places. On a single day, prehospital providers trained a convenience sample of 2253 passers-by in CO-CPR.	Bystander CPR training was not associated with bystander CPR frequency (*β* −0.002; 95% CI −0.16, 0.15), compression- only CPR (*β* −0.06; 95% CI −0.15, 0.02), ROSC (*β* −0.06;95% CI −0.21, 0.25), survival (*β* −0.02; 95% CI −0.11, 0.06), or favourable neurologic outcome (*β* −0.01; 95% CI −0.07, 0.09).	**Training** Instructor-led. CO-CPR. One shot, not targeted initiative.
Nielsen 2012 [21]	Prospective cohort study Denmark, EU	11679 people trained. 35 witnessed by bystander OHCA	Short 24-min DVD-based-self-instruction BLS courses were offered to laypersons. Information about the enrolment was provided through television announcements. Laypersons could also participate in 4-h BLS/AED courses.	9226 people (22% of the population) completed the short course and 2453 (6% of the population) completed the 4-h course. For the witnessed OHCAs (N = 35) the bystander BLS rate increased [22% vs. 74% (95% CI 58–86). No change in survival to hospital discharge [11% (95% CI 4–27)]	**Training** Instructor-led and self-training with video. CO-CPR. TV announcements. Not targeted initiative.
Nielsen 2014 [22]	Prospective cohort study Denmark, EU	124 patients with OHCA in the follow up and 90 in the intervention period.	1. 24-min DVD-based-self-instruction BLS courses. 2. 4-h BLS/AED courses. 3. The local television station had approximately 50 broadcasts about resuscitation	Improvement in bystander CPR rate [70% (95% CI 61–77) vs. 47% (95% CI 37–57), *p* = 0.001]. No difference in the 30-day survival [6.7% (95% CI 3–13) vs. 4.6% (95% CI 1–12), *p* = 0.76].	**Training** Instructor-led and self-training with video. CO-CPR. TV announcements. Not targeted inititaitive.
Isbye 2007 [23]	Prospective cohort study Denmark, EU	1877 OHCAs. Population trained: 35 002 at 806 primary Schools.	Instructor-led (School: first tier) and peer to peer (Family: second tier) training.	Population trained: mean, 2.5 persons per pupil; 95% CI 2.4–2.5 Bystander CPR: not improved. (25.0% vs. 27.9%; *p* = 0.16)	**Training** Instructor-led and peer-to-peer. One shot, two tiers initiative. School and families targeted.
Tay 2019 [24]	Prospective Before-after study Singapore	1241 OHCA, 880 before, 361 after. Close to 30,000 individuals were trained in CPR	The Save-A-life (SAL) initiative offered free training in chest-compression only cardiopulmonary resuscitation (CPR) and automated external defibrillator (AED) use, with signups conducted through the local community centres and schools by different agencies, with standardized teaching material.	Higher survival (3.3% vs. 2.2% *p* = 0.23), pre-hospital return of spontaneous circulation (ROSC) (9.1% vs. 5.1% *p* = 0.01), bystander CPR (63.7% vs. 44.8% *p* < 0.001). After adjusting: increased odds ratio (OR) for survival (OR 2.39 [1.02–5.62]), pre-hospital ROSC (OR 1.94 [1.15–3.25]) and bystander CPR (OR 2.29 [1.77–2.96]).	**Training** Instructor led. CO-CPR. Program targeted on schools and community.
**Mass-Media *n* = 1**					
Becker 1999 [25]	Non-randomized controlled study Seattle, USA	2075 OHCAs, 1786 in the “before” period and 289 in the “during” period. 1099 in the intervention communities and 976 in the comparison communities.	Two 30-s Public Service Announcements (PSA) demonstrating CPR for 8 months. Each featured an older couple with the husband experiencing a witnessed cardiac arrest at home and the wife calling 911 and initiating CPR.	Increased bystander CPR rate (43% vs. 55%, *p* < 0.05). The rate remained at 33% in the comparison community (*p* = 0.967)	**Media announcement** Advantage of using a control group.
**Bundled Interventions *n* = 7**					
Wissenberg 2013 [26]	Prospective cohort study Denmark	A study population of 19,468 OHCA patients.	Bundle intervention 1. Mandatory education in resuscitation in elementary schools (January 2005) 2. New guidelines for resuscitation (November 2005) 3.Mandatory resuscitation course when acquiring a driver’s license (October 2006)	Increased bystander CPR (21.1% vs. 40.9% *p* < 0.001)	**Training** Mandatory education in resuscitation in elementary schools and at driver’s license
Ro 2016 [27]	Cross-sectional study Republic of Korea	228,921 responders from 253 counties. 29,052 OHCAs. 4 quartiles Q1 (lower level of capacity) to Q4 (highest level of capacities).	CPR training programs were developed in the early 2000s. The recent guideline for layperson CPR was released in 2011, which outlines 1-h layperson training on CO-CPR, 1.5 to 2 h of first responder training on chest compression with rescue ventilation CPR, and advanced cardiovascular life support training for professional providers. Enforcement of the EMS Act requires mandatory training of all first responders.	Bystander CPR. Of 29,052 OHCA patients with presumed cardiac origin, 11,079 (38.1%) received bystander CPR. Bystander CPR in Q1(lower level of CPR capacity) = 33.9% vs. Q4 (higher level of CPR capacity) 39.4% (*p* < 0.01)	**Training** 1-h layperson training on CO-CPR
Hwang 2017 [28]	Prospective Before-after study Republic of Korea	581 OHCA, divided into three period groups: before (2009–2010) transition (2011) and after (2012–2013)	The university hospital developed the system-wide CPR program for OHCA patients which included interventions at prehospital and hospital levels. CPR education sessions were conducted at public sites. CO-CPR, in addition to standard basic life support techniques, was taught to citizens in schools and workplaces.	CPR education: 1760 people in 2009, 3394 in 2010, 682 in 2011, 3659 in 2012, and 5994 in 2013. Increased bystander CPR rate (without dispatcher assistance) (13.2% vs. 27.7% (*p* value not reported).	**Training** Both CO-CPR and standard CPR education sessions were conducted at public sites, in schools and workplaces
Ro 2019 [29]	Cross-sectional study Republic of Korea	81,250 OHCAs in 254 counties. 228,452 participants responded to the survey of 247 items Classification in quartiles: the highest (Q1), higher (Q2), lower (Q3), and lowest (Q4) counties.	Public CPR campaigns and training for laypersons	Bystander CPR: Q1 63.8% vs. Q4 60.1, OR 1.16 (1.04–1.29) AOR 1.29 (1.13–1.48).	**Training and public campaigns**
Nishiyama 2019 [30]	Prospective cohort study Japan	57,173 residents (14.7%) completed the chest compression–only CPR training and 32,423 (8.3%) completed conventional CPR training. 722 patients with OHCA were eligible for the analysis	1. The Toyonaka City Fire Department has provided a conventional 3-h CPR training consisting of chest compressions, rescue breathing, and AED use and an instructor training course to the residents at companies, governmental offices, and nursing homes. 2. A video-based CPR training program. 3. The participants used a Mr. PUSH CPR training kit to practice chest compressions and AED use. 4. Especially for schools, the Toyonaka City Fire Department introduced systematic training programs with CO-CPR, collaborating municipal board of education.	Proportion of TP:23% of the residents. No increase in bystander CPR: 43.3% in 2010 vs. 42.0% in 2015 (*p* = 0.915). Increase in high-quality CPR 11.7% in 2010 vs. 20.7% in 2015 (*p* = 0.015). No difference in 1-month survival (AOR, 0.949; 95% CI, 0.802–1.124) and 1-month survival with favourable neurological outcome (AOR, 0.947; 95% CI, 0.751–1.194)	**Training** Both CO-CPR and standard CPR. Conventional 3-h CPR training at companies, governmental offices, and nursing homes. A video-based CPR training program. Mr. PUSH CPR training kit.
Kim 2019 [31]	Retrospective, Before-After Study Republic of Korea	1155 OHCAs, 777 from the pre-intervention period and 378 from the post-intervention period	“Train the trainer” instruction to EMS dispatchers who are responsible for instructing bystanders in CPR. CO-CPR training sessions for laypersons. The Korean Society of EMS Physicians performed lectures for dispatchers and instituted regular review of dispatch records. Dispatchers conducted the CO-CPR trainings for first responders, such as police officials, as well as laypersons. Korea University Ansan Hospital instituted regular skills training sessions for EMTs in that service area. A detailed data collection instrument to be completed by EMTs for each cardiac arrest.	Bystander CPR before and after intervention 13.2% vs. 37.4% (risk difference [RD] 24.2%; 95% CI, 18.2%–29.4%). No significant improvement (*p* value not available)	**Training** CO-CPR
Cone 2020 [32]	Retrospective cohort USA	HEARTSafe-designated communities and non-designated communities. 2922 SCA cases (CARES): 1569 (54%) occurred in towns that were HEARTSafe-designated	CPR training, availability of automated external defibrillators (AEDs) on first responder vehicles and through public access defibrillation initiatives, and availability of post-arrest therapeutic hypothermia and percutaneous coronary intervention at receiving hospitals.	No improvement in Bystander CPR. Lay person 399 (25.45%) in HEART Safe communities vs. 337 (24.91%) in non HEART Safe. CPR performed by bystander vs other. Unadjusted OR 1.019, (95% CI 0.814, 1.275), *p* = 0.8722 Adjusted OR 1.147, (95% CI 0.893, 1.473), *p* = 0.2838	**Training**

**Table 2 jcm-10-05719-t002:** The impact of each intervention on the assessed outcomes.

Intervention	Outcomes [Reference]									
	Survival to Hospital Discharge with Good Neurological Outcome		Survival to Hospital Discharge		ROSC		Bystander CPR Rate		Proportion of Trained People	
Mass-Media Campaigns							1 [25]			
Community Training Programms (+/− Media)	2 [15], [16] *	2 [16] **, [20]	2 [16] ***, [24]	4 [18,20,21,22]	1 [24]	1 [20]	7 [15], [16] ***, [17,18,21,22,24]	3 [14,20,23]	3 [17,19,23]	-
Bundle Interventions	-	1 [30]	-	1 [30]	-	-	4 [26,27,28,29]	3 [30,31,32]	2 [28,30]	
								**Intervention Influence on Outcome**		
								**Yes**	**No**	**No Data Reported**


* Only in public. ** Only at home. *** In public and at home.

## Data Availability

Not applicable.

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
