# Peer review of "Community Initiatives to Promote Basic Life Support Implementation—A Scoping Review"

_jcm, 2021, doi:10.3390/jcm10245719_

Round 1

Reviewer 1 Report

The report is generally well-written. One pervasive problem is lack of consistency in terminology. Compression only cardiopulmonary resuscitation is referred to variously as CO CPR, CO-CPR, hands-only compression CPR, CCCPR, and Hands-only CPR (at least). It is important to distinguish this from standard CPR. It is confusing to use a variety of expressions for it.

A few minor grammatical issues and needs for clarification are identified in the attached file.

Author Response

Thank you for your comments and suggestions:

  • We accepted all your corrections in the text of the manuscript including unclear figures in the results section;
  • We corrected the table:
    • by using "CO-CPR" for chest compressions-only CPR;
    • deleting the exceeding reference numbers in the first column;
    • rewording comments in the last column to get them more uniform. 

Please, see the attachment for your evaluation.

Regards,

The authors

Reviewer 2 Report

The topics covered are essential to the practice of resuscitation. In my opinion, an element that will significantly improve patient survival in the future is cooperation with the public, social initiatives, legal initiatives, the use of modern teaching techniques and modern technology, including drones, smartphones, and other electronic devices. I believe that in the next few years we will see a revolution in this area. 
This manuscript submitted for review is a very interesting paper from a practical point of view, and it addresses current issues that were mentioned in detail in the ERC 2021 Guidelines. Despite the limitations of the methodology adopted, this analysis shows the current state of knowledge and informs the direction of further research and related needs. Of primary importance is the support and appreciation of community initiatives and the importance of the impact of specific interventions involving lay communities on bystander BLS rates and other consistent clinical outcomes, and to identify relevant knowledge gaps.  
The authors correctly defined the methodology of the study performed the analysis in a methodologically correct manner with an analysis of the risk of methodological errors and limitations due to the methodology. They presented rigorously the strategy for finding and analyzing scientific evidence. They presented the methodology and purpose of the study at the highest level. They correctly analyzed the results obtained and presented an interesting discussion of the results obtained. 
I have no objection to the ethical aspects of the work presented. The authors' conflict of interest was correctly presented.
Author Response

Thank you very much for your comment and appreciation.